# Immune Escape Is an Early Event in Pre-Invasive Lesions of Lung Squamous Cell Carcinoma

**DOI:** 10.3390/diagnostics10070503

**Published:** 2020-07-21

**Authors:** David Laville, Francois Casteillo, Violaine Yvorel, Olivier Tiffet, Jean-Michel Vergnon, Michel Péoc’h, Fabien Forest

**Affiliations:** 1Pathology Department, North Hospital, University Hospital of Saint Etienne, Avenue Albert Raimond, CEDEX 2, 42055 Saint Etienne, France; davlaville@gmail.com (D.L.); francois.casteillo@chu-st-etienne.fr (F.C.); violaine.yvorel@chu-st-etienne.fr (V.Y.); michel.peoch@chu-st-etienne.fr (M.P.); 2Thoracic Surgery Department, North Hospital, University Hospital of Saint Etienne, Avenue Albert Raimond, CEDEX 2, 42055 Saint Etienne, France; olivier.tiffet@chu-st-etienne.fr; 3Pneumology Department, North Hospital, University Hospital of Saint Etienne, Avenue Albert Raimond, CEDEX 2, 42055 Saint Etienne, France; j.michel.vergnon@chu-st-etienne.fr; 4Corneal Graft Biology, Engineering and Imaging Laboratory, BiiGC, EA2521, Federative Institute of Research in Sciences and Health Engineering, Faculty of Medicine, Jean Monnet University, 42000 Saint-Etienne, France

**Keywords:** bronchial dysplasia, PD-L1, CD8, immunotherapy

## Abstract

Bronchial dysplasia is the pre-neoplastic lesion recognized for invasive squamous cell carcinoma. The mechanisms leading to invasive squamous cell carcinoma for this lesion are not fully known. Programmed Death-Ligand 1 (PD-L1) expression by the bronchial dysplasia neoplastic epithelium might suggest a response to immunotherapy. The objective of this work is to further characterize PD-L1 and CD8 expression in bronchial dysplasia and bronchial metaplasia compared to normal bronchial epithelium. Immunohistochemical analysis of PD-L1 and CD8 staining were characterized in bronchial dysplasia of 24 patients and correlated with clinical data. We also compared PD-L1 expression in dysplasia samples to 30 normal epithelium and 20 samples with squamous bronchial metaplasia. PD-L1 was never expressed in normal epithelium and in metaplastic epithelium whereas 37.5% of patients with bronchial dysplasia were stained by PD-L1 (*p* < 0.001). PD-L1 expression was not related to the degree of dysplasia or a medical history of invasive squamous cell carcinoma, while CD8 expression and its localization were related to medical history of squamous cell carcinoma (*p* = 0.044). Our results show that PD-L1 is expressed in roughly one third of patients with bronchial dysplasia and is not expressed in normal and metaplastic epithelium. This suggests that PD-L1 is expressed in preneoplastic lesions of squamous cell carcinoma.

## 1. Introduction

Lung cancer is the leading cause of cancer-related deaths worldwide. For bronchial and lung neoplasms, three pre-invasive lesions are recognized: (1) bronchial dysplasia for lung squamous cell carcinoma, (2) atypical adenomatous hyperplasia for lung adenocarcinoma and (3) diffuse idiopathic pulmonary neuroendocrine hyperplasia for carcinoids. Bronchial dysplasia can arise from any part of the bronchial tree, predominantly in tobacco smokers. Bronchial dysplasia is graded as mild, moderate or high grade dysplasia with a higher risk of progression to invasive squamous carcinoma for patients with high grade dysplasia. Endoscopic imaging with fluorescence endoscopy has enhanced the detection and location of bronchial dysplasia, being now mandatory for patients’ management. Therapeutic options are discussed for each patient according to the grade of dysplasia and the extent of the lesions. Most bronchial dysplasia lesions will not progress to invasive carcinoma but the persistence of dysplasia can lead to invasive cancer especially in high risk lesions [1,2]. Available therapies are multiple and include smoking cessation, minimally invasive treatment such as endoscopic therapy, and follow-up. The main goal of these therapies is to prevent the evolution to squamous cell carcinoma. For some patients, bronchial dysplasia lesions do not regress after smoking cessation or local endoscopic treatment, or bronchial dysplasia lesions are widely spread through the bronchial tree and cannot be treated by endoscopic therapy. There is no current systemic therapy that can prevent the evolution to lung squamous cell carcinoma or lead to an increase of bronchial dysplasia regression. Furthermore, the mechanisms involved in the progression and the recognition of these lesions by the immune system are not fully known.

Programmed Death Ligand-1 (PD-L1) is a membranous protein expressed by tumor cells that can bind to PD1 expressed by lymphocytes to modulate the immune system; this reduces both proliferation and activity of cytotoxic CD8 T cell response to cancer associated antigens, and may allow tumor cells to avoid immune surveillance. In lung cancer, several immunotherapies targeting PD1/PD-L1 are approved: durvalumab, atezolizumab, nivolumab and pembrolizumab. Positive PD-L1 expression is required for pembrolizumab (stage IV) and durvalumab (unresectable stage III in some countries/regions) in not only squamous cell carcinoma but also non-squamous cell carcinoma [3,4,5].

For bronchial dysplasia, little is known about its microenvironment and the expression of PD-L1 that could represent a potential therapeutic target. A recent study showed that inflammatory pathway activity is changed in high-risk persistent bronchial dysplasia and may be related to progression to invasive squamous cell carcinoma [6]. Another more recent study showed that PD-L1 expression is increased in high grade dysplasia [7]. Similarly, PD-L1 expression has been found at various levels in other pre-invasive epithelial lesions in squamous cell epithelium lesions such as intraepithelial neoplasia of the cervix and preinvasive oral lesions [8,9].

Because PD-L1 is a common mechanism of immune escape in invasive squamous cell carcinoma, we wondered if bronchial dysplasia already expresses this protein at an early state of carcinogenesis.

In this exploratory study, we analyzed the expression of PD-L1 and CD8 in bronchial biopsies and surgical samples from 24 patients with bronchial dysplasia consecutively gathered. We analyzed (1) the relationship between PD-L1 and CD8 expression with the degree of dysplasia, (2) the relationship between PD-L1 or CD8 expression and medical history of invasive squamous cell carcinoma, (3) and the proportion of PD-L1 positive lesions of bronchial dysplasia.

## 2. Materials and Methods

### 2.1. Subjects and Tissue Samples

This study was performed on patients previously diagnosed with bronchial dysplasia. Medical information was taken from medical records. The Ethics Committee of the University Hospital of Saint Etienne (Institutional Review Board: IORG0007394) approved this study, IRBN272019/CHUSTE approved on 1 April 2019. Paraffin-embedded, formalin-fixed (FFPE) tissue was available from our records at the North Hospital, Saint Etienne. We included biopsy samples, endobronchial resection samples and surgical specimens gathered between 2010 and 2018, when the main diagnosis retained was dysplasia or in situ carcinoma, regardless of the degree of consecutive dysplasia. Cases were re-evaluated jointly by a sub-specialty-trained thoracic pathologist (FF) and a pathology fellow (DL) and classified according to the current 2015 World Health Organization (WHO) classification. We excluded cases where available tumor material was not available to realize further immunohistochemistry study. Cases with less than 100 dysplastic cells were excluded. Cases with concomitant invasive squamous cell carcinoma and dysplasia at the edge of invasive squamous cell carcinoma were excluded.

FFPE slides were stained with Hematoxylin & Eosin for subsequent morphologic evaluation and classification. Bronchial tissue was classified into one of the four histologic categories as defined by the 2015 WHO classification of thoracic neoplasms, as follows: 1: mild grade dysplasia, 2: moderate grade dysplasia, 3: high grade dysplasia and 4: in situ carcinoma. Grouping of these diagnoses into Low Grade Lesions (mild and moderate grade dysplasia) and High Grade Lesions (high grade dysplasia and carcinoma in situ) categories are used for subgroup study.

We compared the results of PD-L1 expression obtained in bronchial dysplasia to PD-L1 expression in 30 cases of normal bronchial epithelium and to 20 cases of squamous metaplasia without atypia.

### 2.2. Immunohistochemistry

Automated immunohistochemistry was performed on Omnis platform (Agilent, USA). The following antibodies were used: PD-L1 (22C3, dilution 1/40, Dako-Agilent) and CD8 (C8/144B, dilution 1/100, Dako-Agilent). Appropriate external (tonsil) and internal controls (mononuclear cells) were used.

### 2.3. Assessment of Immunohistochemical Staining

PD-L1 expression was quantified by two observers jointly (DL &FF) evaluating the percentage of neoplastic cells with membranous expression at x200 magnification. The percentage of neoplastic stained cells was evaluated on the whole slide. We also assessed PD-L1 expression in immune cells and the localization of immune cells, within the neoplastic epithelium or within the submucosa below the dysplastic epithelium. CD8 expressing cells were characterized by their localization: within the dysplastic epithelium, in the sub-mucosal tissue, in both or absent. The percentage of CD8 positive lymphocytes were evaluated. The density of CD8 positive cells in the submucosa and in the epithelium were also evaluated on scanned slides (Aperio, Leica Biosystems, Vista, CA, USA) with QuPath Software (version 0.2.1, University of Edinburgh, Edinbugh, UK, available at https://qupath.github.io).

### 2.4. Statistical Analysis

Statistical analysis was performed with R software for Linux (version 3.2.3, R Core Team, Vienna, Austria, available at https://www.R-project.org/) and Rstudio for Linux (Version 1.0.143, RStudio Team. RStudio: Integrated Development for R. RStudio, Boston, MA). Descriptive statistics such as mean and standard deviation for continuous variables are provided. The primary analyses focused on the relationship between histologic grade and the expression of PD-L1 and CD8. Theses analyses were based on every tissue sample and realized with the Exact Test of Fisher for categorical variable. Pearson’s test was used for quantitative variables. Results are reported as two-sided *P*s and/or 95% confidence intervals. Failed or missing data are not included in statistical analysis. 

Cohen’s weighted kappa was calculated to assess the reproducibility in the two observers between categorical variables. The Shapiro-Wilk test was used to test normality for continuous variables. To assess the relationship between continuous variables when the normality was not present Spearman test was used and ρ are given; the Pearson correlation test was used when normality was present and R2 are given. The Wilcoxon-Mann-Whitney test was used to compare means of continuous variables.

## 3. Results

### 3.1. Patients and Tissue Samples

Within the inclusion period, 24 tissue samples from 24 patients with bronchial dysplasia were retained, having enough material to perform immunohistochemistry. Most patients were males (23/24, 95.8%) with a mean age of 64+/−8 years old. All patients with dysplasia were smokers with a mean pack-years of 44.3+/−5.5. Samples were represented by bronchial biopsies (*n* = 19), surgical resections (*n* = 4) and endobronchial resections (*n* = 1).

None of our patients had mild-grade dysplasia, 10 patients had moderate dysplasia, nine severe dysplasia and five were graded as in situ carcinoma.

### 3.2. PD-L1 in Dysplasia and Comparison with Normal Epithelium and Metaplasia

PD-L1 was never expressed in normal epithelium and in metaplastic epithelium whereas nine (37.5%) patients with bronchial dysplasia were stained by PD-L1 (*p* < 0.001). The mean percentage of stained dysplastic cells by PD-L1 was 15+/−5.7%. Spearman’s test ρ for the evaluation of the relationship between the percentage of dysplastic cells stained by PD-L1 between the two observers was at 1.

These results are summarized in Table 1 and illustrative images are shown in Figure 1. 

### 3.3. PD-L1 and CD8 Evaluation in Dysplasia

For dysplasia, CD8 positive lymphocytes were found in the submucosa in 20 (83.3%) patients and in 11 (32.3%) patients within dysplastic epithelial cells (Table 2). Between two observers, Cohen’s weighted kappa for the presence of the lymphocytes stained by CD8 was at 0.68 for submucosal localization, and 0.73 for the localization within epithelial cells. The mean percentage of lymphoid cells stained by anti-CD8 was 26+/−4.1%. Spearman’s test ρ for the evaluation of the relationship between the two observers for the percentage of lymphocytes stained by anti-CD8 between the two observers was at 0.81.

### 3.4. PD-L1 and CD8 Expression and Degree of Dysplasia

The results are summarized in Table 1. Severe dysplasia and carcinoma in situ are grouped for statistical analysis. There was no significant difference in PD-L1 expression by dysplastic cells in both groups (*p* = 1). PD-L1 expression by immune cells located within the neoplastic epithelium and in the sub-mucosa was not related to the degree of dysplasia (*p* = 0.68 and 0.20 respectively). There was no association between CD8 expression in the different localizations and the degree of dysplasia (*p* = 0.78). There was no relationship between the degree of dysplasia and the number of CD8+ cells/mm^2^ (*p* = 0.84).

### 3.5. P D-L1 and CD8 Expression and Medical History of Squamous Cell Carcinoma

The main data related to PD-L1 and CD8 expression and the medical history of squamous cell carcinoma are summarized in Table 2. Twelve patients had a previous or a following history of bronchial invasive squamous cell carcinoma. CD8 expression and its localization were related to the medical history of squamous cell carcinoma (*p* = 0.044). When a medical history of squamous cell carcinoma was present, CD8 positive cells were present in the epithelium and the sub-mucosa in 33% of patients whereas it was present in only 8% of patients without medical history of squamous cell carcinoma. Using image analysis with Qupath software, the mean density of lymphocytes was at 1503/mm^2^+/−231 in the submucosa. Mean density of CD8 positive cells in the submucosa was measured at 1693/mm^2^+/−334 and 1384/mm^2^+/−322 lymphocytes in patients with and without a medical history of squamous cell carcinoma, respectively (*p* = 0.72). Mean density of CD8 positive cells within the epithelium is 674/mm^2^+/−251. Mean density of CD8 positive cells in the epithelium was measured at 703/mm2+/−342 and 625/mm2+/−407 lymphocytes in patients with and without a medical history of squamous cell carcinoma, respectively (*p* = 1). PD-L1 expression was not related to the medical history of squamous cell carcinoma.

## 4. Discussion

PD-L1 expression by neoplastic cells of bronchial dysplasia has been suggested by recently published reports. On immortalized *KRAS* mutated bronchial epithelial cells, it has been shown that PD-L1 mRNA levels were higher than control and might be enhanced by LKB1 (Liver Kinase B1) loss and mediated by MAPK/ERK (Mitogen-Activated Protein Kinases/Extracellular signal-Regulated Kinases) pathway activation [10]. It has also been suggested that there is a depletion of innate and adaptive immune cells in non-regressive/proliferative lesions compared with regressive lesions [11]. Another recent work described the genomic, epigenomic and transcriptomic alterations and provides a molecular map of bronchial dysplasia [12]. The factors determining the fate of bronchial dysplasia, persistence, regression, or progression as an invasive squamous cell carcinoma, are still not fully known.

Some patients with bronchial dysplasia are refractory to existing therapies and might represent candidates for treatment with checkpoint inhibitors. However, the usefulness of an immunotherapeutic approach has not been demonstrated in such pre-invasive lesions. We evaluated PD-L1 and CD8 expression in bronchial dysplasia. The primary findings of our work are: (1) that PD-L1 can be expressed by dysplastic epithelium and is not expressed in bronchial metaplasia and normal epithelium, (2) a subset of dysplastic epithelium express PD-L1, and (3) the presence and topography of CD8 expressing T-cells might be related to a medical history of invasive squamous cell carcinoma.

On precancerous lesions, PD-L1 expression has only recently been infrequently studied in preinvasive lesions of various organs. These studies are not comparable mostly because the PD-L1 antibody used for immunohistochemistry is not the same. Another bias making these studies not comparable is that the carcinogenesis of each lesion is different: microsatellite instability for serrated lesions of the colon, viral induced carcinogenesis for cervical intra-epithelial neoplasia and tobacco-induced carcinogenesis for bronchial dysplasia [9,13,14,15]. The choice of PD-L1 clone is crucial because all clones are not interchangeable [16,17]. The cut-off for positivity and methods for PD-L1 stain evaluation vary among studies with some only evaluating the percentage of stained neoplastic cells, while others evaluate the intensity of staining and inflammatory cell staining. The 22C3 clone chosen for our study is the one used for pembrolizumab eligibility in non-small cell lung cancer [18].

We have shown that roughly one third of patients with bronchial squamous dysplasia express PD-L1. Although limited because of the rarity of this lesion, our study provides new insights into the frequency of PD-L1 expression in all the consecutive bronchial dysplasia in our center. PD-L1 expression is one of the mechanisms involved in immune escape from the microenvironment. This result highlights the fact that PD-L1 expression is an early event in the carcinogenesis of lung squamous cell carcinoma, although this preneoplastic lesion is not directly invading the submucosa. This result follows other studies analyzing PD-L1 expression in pre-invasive lesions arising in the squamous epithelium [8].

Several attempts to find possible therapeutic targets on preneoplastic lesions of bronchial epithelium were performed. A study with 18 lesions considered as bronchial dysplasia showed that none of them had an amplification of *HER2* but that *EGFR* amplification was found in one third of cases [19]. A recent study of gene expression in persistent bronchial dysplasia identified alterations in the inflammatory pathway in persistent bronchial dysplasia associated to invasive squamous cell carcinoma [6].

The persistence of bronchial dysplasia is associated with the development of invasive squamous cell carcinoma [1]. An early detection of lung cancer is associated with better survival as shown in the Nelson study. Nevertheless, bronchial dysplasia cannot be detected by computerized tomography scan. Our work is of importance in this rare lesion, highlighting that a subset of this lesion might be eligible to a general therapy and underlining an immune escape mechanism present in this early lesion.

In this study, the localization of CD8 positive cells is associated with a medical history of invasive squamous cell carcinoma. This morphological result might be the reflection of the modified inflammatory activity in high risk persistent bronchial dysplasia [6]. For dysplasia of the cervix, another dysplasia in squamous cell epithelium but due to viral infection, a study showed that PD-L1 expressing inflammatory cells were located around dysplastic cells and are CD8 positive [9]. In the study on oral precancerous lesion, subepithelial CD163-positive cell count was the only significant risk factor for high-grade dysplasia [20]. Our results follow these findings suggesting that the localization of CD8 positive cells is more frequently found in the epithelium +/− the submucosa; whereas CD8 positive cells are only located in the submucosa when no medical history of squamous cell carcinoma was found. For atypical adenomatous hyperplasia, which is the pre-invasive counterpart of bronchial dysplasia for lung adenocarcinoma, it has recently been shown that tumors positive for both PD-L1 and CD8 had a larger proportion of subclonal mutations [21].

We did not show any correlation between the degree of dysplasia and PD-L1 expression nor with a medical history of invasive squamous cell carcinoma. Our studied population is too small to show a difference because bronchial squamous dysplasia is a rare lesion, and because remaining tissue was not always available. Nevertheless, our study has a comparable size to other works on this type of lesion [2]. Another limit of our study is inherent in its retrospective nature with a limited follow-up. Furthermore, it would have been interesting to test PD-L1 expression in patients with associated subsequent invasive squamous cell carcinoma in negative and positive cases, but most pathological material of invasive tumors was not available. Furthermore, PD-L1 status might not be representative on small samples leading to underestimation of PD-L1 expression rather than overestimation [22].

In conclusion, our work shows that a subset of bronchial dysplasia epithelium expresses PD-L1, underlying the fact that PD-L1 expression is an early event for immune escape in preinvasive lesions of bronchial squamous cell carcinoma. Furthermore, PD-L1 is not expressed in bronchial metaplasia or normal epithelium; this marker might be useful in rare difficult cases to help for the distinction within these entities. Nevertheless, the diagnosis of bronchial dysplasia remains morphological. CD8 expression is related to a medical history of squamous cell carcinoma supporting the fact that high risk severe dysplasia lesion can modify its microenvironment to enhance cancer progression.

## Figures and Tables

**Figure 1 diagnostics-10-00503-f001:**
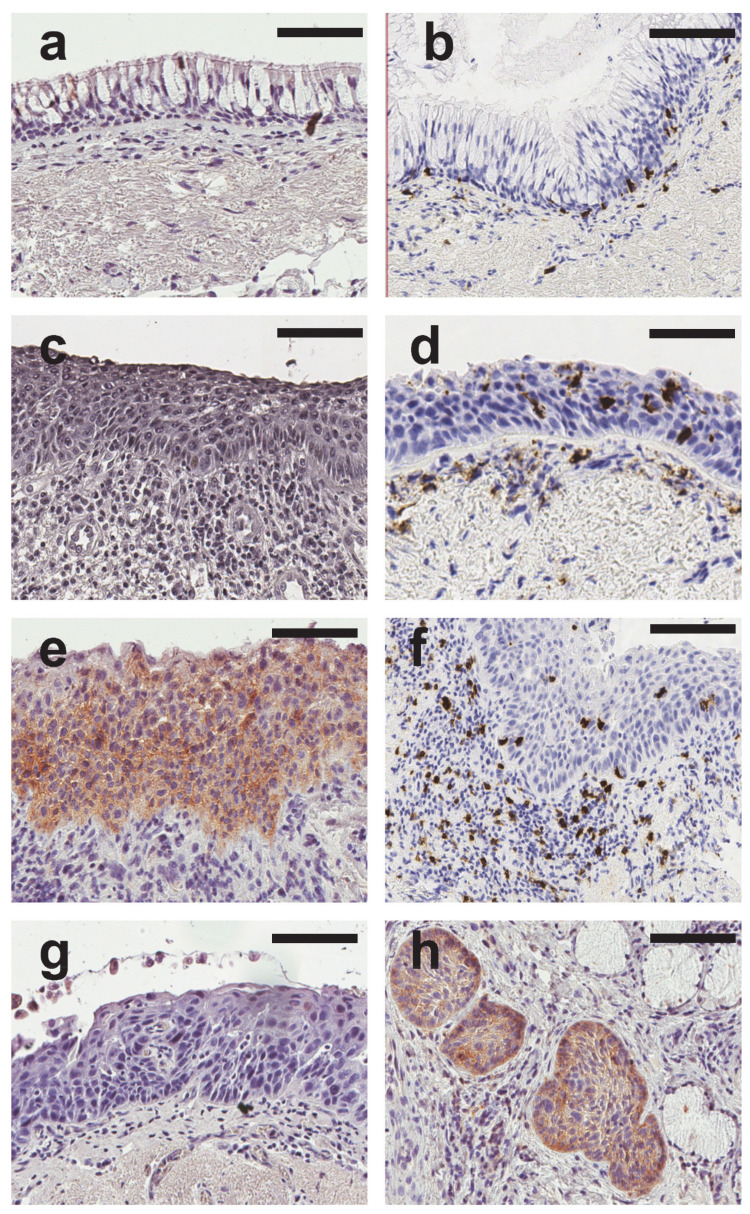
Representative images of PD-L1 and CD8 staining in normal, metaplastic and dysplastic epithelium. All images were taken at ×200 of magnification. Scale bar = 100 µm. (**a**) Immunohistochemistry anti-PD-L1 in normal bronchial epithelium showing that epithelial cells are not stained. (**b**) Immunohistochemistry anti-CD8 in normal bronchial epithelium showing a staining of scattered lymphocytes in the submucosa. (**c**) Immunohistochemistry anti-PD-L1 in bronchial metaplasia showing that epithelial cells are not stained. (**d**) Immunohistochemistry anti-CD8 in bronchial metaplasia showing staining of intraepithelial lymphocytes and lymphocytes in the submucosa. (**e**) Immunohistochemistry anti-PD-L1 in bronchial dysplasia showing a staining of dysplastic cells in this case of moderate dysplasia. (**f**) Immunohistochemistry anti-CD8 in bronchial dysplasia showing staining of intraepithelial lymphocytes and lymphocytes in the submucosa. (**g**) Immunohistochemistry anti-PD-L1 in bronchial dysplasia showing that none of the dysplastic cells are stained. (**h**) bronchial glands colonization by PD-L1 positive dysplastic cells without stromal invasion in a case of high grade dysplasia.

**Table 1 diagnostics-10-00503-t001:** Relationship between Programmed Death-Ligand 1 (PD-L1) and CD8 expression in dysplastic neoplasms.

	PD-L1 Expression by Dysplastic Cells		
	Not Expressed, *n* (%)	Expressed, *n* (%)	*p*
**Age**			0.36
> 60 years old	3 (12.5)	4 (16.7)	
≤ 60 years old	12 (50)	5 (20.8)	
**CD8 expressing cells localization**			0.99
Sub-mucosal	6 (25)	4 (16.7)	
Intra-epithelial	1 (4.2)	0 (0)	
Sub-mucosal and intra-epithelial	6 (25)	4 (16.7)	
No expression	1 (4.2)	0 (0)	
Not assessable	1 (4.2)	1 (4.2)	
**PDL-1 expression by intra-epithelial-immune cells**			0.21
PD-L1 expressed	9 (37.5)	3 (12.5)	
PD-L1 not expressed	5 (20.8)	7 (29.2)	
**PDL-1 expression by sub-mucosal immune cells**			0.65
PD-L1 expressed	4 (16.7)	4 (16.7)	
PD-L1 not expressed	8 (33.3)	4 (16.7)	
Not assessable	2 (8.3)	2 (8.3)	

**Table 2 diagnostics-10-00503-t002:** Relationship between dysplasia groups, medical history of squamous cell carcinoma and PD-L1, and CD8.

	Mean Age	Pack-Years	Moderate Dysplasia, n (%)	Severe Dysplasia, Carcinoma in Situ, n (%)	*p*	No Medical History of Squamous Cell Carcinoma, n (%)	Medical History of Squamous Cell Carcinoma, n (%)	Unknown, n (%)	*p*
**PD-L1 expression by dysplastic cells**					1				1
0%	64+/−3	46+/−7	6 (25)	9 (37)		6 (25)	8 (33)	1 (4)	
≥ 1%	63+/−6	41+/−8	4 (17)	5 (21)		4 (17)	4 (17)	1 (4)	
**PD-L1 expression by intraepithelial immune cells**					0.68				0.67
Not expressed			6 (25)	6 (25)		6 (25)	5 (21)	0 (0)	
Expressed	62+/−3	47+/−8	4 (17)	8 (33)		4 (17)	7 (17)	0 (0)	
**PD-L1 expression by sub-mucosal immune cells**	65+/−2	44+/−8			0.20				0.65
Not expressed	68+/−1	36+/−7	7 (29)	5 (21)		4 (17)	7 (29)		
Expressed	62+/−3	53+/−1	2 (8)	6 (25)		4 (17)	4 (17)		
Not assessable	60	30	1 (4)	3 (12)		2 (8)	1 (4)	1	
**Previous or following squamous cell carcinoma**					0.66				
No	66+/−2	61+/−9	5 (21)	5 (21)					
Yes	62+/−3	33+/−5	4 (17)	8 (33)					
Unknown	62+/−2	25+/−5	1 (4)	1 (4)					
**CD8 expressing cells localization**					0.78				**0.044**
Sub-mucosal	64+/−2	55+/−8	4 (17)	6 (25)		6 (25)	3 (12)	1 (4)	
Intra-epithelial	65	20	1 (4)	0 (0)		0 (0)	0 (0)	1 (4)	
Sub-mucosal and intra-epithelial	67+/−2	33+/−8	4 (17)	6 (25)		2 (8)	8 (33)	0 (0)	
No expression	61	40	0	1 (4)		1 (4)	0 (0)	0 (0)	
Not assessable	48+/−10	65+/−15	1 (4)	1 (4)		1 (4)	1 (4)	0 (0)	

No relationship was found between the percentage of PD-L1 neoplastic cells and the percentage of CD8 positive immune cells (*p* = 0.44). PD-L1 expression by tumor cells was not associated with PD-L1 staining of intra-epithelial or sub-mucosal immune cells by this antibody (*p* = 0.21 and *p* = 0.65 respectively). The number of pack-years was not related to PD-L1 expression and CD8 positive lymphocytes localization (*p* = 1). Bold: significant *p*-values.

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
