# Peer review of "Immune Escape Is an Early Event in Pre-Invasive Lesions of Lung Squamous Cell Carcinoma"

_diagnostics, 2020, doi:10.3390/diagnostics10070503_

Round 1

Reviewer 1 Report

This study evaluates the association of PDL1 and CD8 positive cells with degree of preneoplastic atypia and the presence of associated lung squamous cell carcinoma (SCC) history (prevalent and incident) in bronchial dysplasia (BD).  Immunohistochemical stains are used to detect and quantify the positive cells in the epithelial compartment (PDL1 positive epithelial and inflammatory cells and CD8 positive T-lymphocytes) and submucosal compartment (CD8 and PDL1 positive inflammatory cells).  It appears that lesions are scored as positive if any positive cells are present.  The key findings are the association of PDL1 positivity with presence of dysplastic epithelial change as compared to normal bronchial epithelium and the association of increased CD8 positive LC presence in lesions from patients with lung SCC history versus those without.  These findings have not been directly demonstrated in situ in bronchial dysplasia previously and parallel similar findings in other premalignant lesions thus providing informative data.

The study focuses on independent BD that meet a microscopic size cutoff giving a relatively pure study set.  The small number of cases included is a limitation of the study and a number of other weaknesses are noted related to methodology and statistical comparisons.  Given the small size of the study set and the inclusion of both prevalent and incident SCC cases in that carcinoma positive cohort (i.e. SCC prior to and after the studied BD, respectively), it seems that inclusion of BD collected concomitantly with the identification of an invasive SCC would be helpful.  Excluding BD that is contiguous with (“at the edge of”) invasive SCC is appropriate as it may represent cancerization of the airways, but if additional cases of independent BD from concomitant SCC cases could be included this could help improve power.  A methodological concern is the lack of a description of how the denominator for the percent of CD8 positive cells is calculated.  It appears this may simply be a count of cells histologically considered to be consistent with lymphocytes on H&E stained slides.  A more appropriate count would be based on overall CD45 positive cell numbers as the denominator.  In lieu of this, a comparison of reproducibility of counts by the two scorers would be helpful to provide a level of confidence in the accuracy of the quantification.  Regardless, analyses based on percent of CD8 positive LCs and/or raw numbers per area of BD or associated underlying submucosa would be worthy of evaluation.  The increase of CD8 cells in poorer prognosis cases presumably correlates to an inactive status of these cells.  The demonstration of an increase in density and/or proportion of these cells in the invasive SCC group would provide some indication of the degree of unresponsiveness in these settings.  Conversely, the dichotomous classification of PDL1 positive and negative cases is common and parallels scoring for invasive lung cancers.  However, data should be presented regarding the statistically significant difference in PDL1 positive BD cases versus the normal cohort (does this cohort include squamous metaplasia without atypia?).  It is not too surprising that there are no detectable differences between moderate dysplasias and the high grade group consisting of severe BD and CIS as moderate dysplasias show characteristics that are often closer to high grade than low grade subgroups.  Thus, presenting the data differentiating the BD from the normal tissues should be highlighted.

While the small size of the study set and the limitations of IHC as the only technique to quantify cell populations are drawbacks, with some major revisions the manuscript could provide some informative observations related to the progression of premalignant lesions to invasive lung cancer.

Author Response

Reviewer#1

We would like to thank reviewer#1 for her/his comments. We have chosen to resubmit our revised manuscript with the revision suggested by reviewers. We think that these revisions have improved the manuscript and hope you will find it suitable for publication.

This study evaluates the association of PDL1 and CD8 positive cells with degree of preneoplastic atypia and the presence of associated lung squamous cell carcinoma (SCC) history (prevalent and incident) in bronchial dysplasia (BD).  Immunohistochemical stains are used to detect and quantify the positive cells in the epithelial compartment (PDL1 positive epithelial and inflammatory cells and CD8 positive T-lymphocytes) and submucosal compartment (CD8 and PDL1 positive inflammatory cells).  It appears that lesions are scored as positive if any positive cells are present.  The key findings are the association of PDL1 positivity with presence of dysplastic epithelial change as compared to normal bronchial epithelium and the association of increased CD8 positive LC presence in lesions from patients with lung SCC history versus those without.  These findings have not been directly demonstrated in situ in bronchial dysplasia previously and parallel similar findings in other premalignant lesions thus providing informative data.

The study focuses on independent BD that meet a microscopic size cutoff giving a relatively pure study set.  The small number of cases included is a limitation of the study and a number of other weaknesses are noted related to methodology and statistical comparisons.  Given the small size of the study set and the inclusion of both prevalent and incident SCC cases in that carcinoma positive cohort (i.e. SCC prior to and after the studied BD, respectively), it seems that inclusion of BD collected concomitantly with the identification of an invasive SCC would be helpful.  Excluding BD that is contiguous with (“at the edge of”) invasive SCC is appropriate as it may represent cancerization of the airways, but if additional cases of independent BD from concomitant SCC cases could be included this could help improve power.-> We do not wish to include BD at the edge of SCC because we cannot exclude that BD at the edge of SCC might be a superficial spread of SCC and might not be BD in fact. We wish our case series to be made of true bronchial dysplasia.

A methodological concern is the lack of a description of how the denominator for the percent of CD8 positive cells is calculated.  It appears this may simply be a count of cells histologically considered to be consistent with lymphocytes on H&E stained slides.  A more appropriate count would be based on overall CD45 positive cell numbers as the denominator. In lieu of this, a comparison of reproducibility of counts by the two scorers would be helpful to provide a level of confidence in the accuracy of the quantification.  -> The slides were evaluated by two pathologists for the concordance of PD-L1 and CD8 expression. We have added the following sentences in the manuscript :

  • P6, l42-144. : Spearman’s test ρ for the evaluation of the relationship between the percentage of dysplastic cells stained by PD-L1 between the two observers was at 1.
  • P6, l173-175 : Between two observers, Cohen’s weighted kappa for the presence of the lymphocytes stained by CD8 was at 0.68 for submucosal localization, and 0.73 for the localization within epithelial cells..
  • P6, l177-179 : in paragraph 3.2 : Spearman’s test ρ for the evaluation of the relationship between the two observers for the percentage of lymphocytes stained by anti-CD8 between the two observers was at 0.81.

We have also added the following sentences in p3, l24-129 : Cohen’s weighted kappa was calculated to assess the reproducibility between the two observers between categorical variables. Shapiro-Wilk test was used to test normality for continuous variables. To assess the relationship between continuous variables when the normality was not present Spearman test was used and ρ are given, and Pearson correlation test when normality was present and R2 are given. Wilcoxon-Mann-Whitney test was used to compare means of continuous variables.

Regardless, analyses based on percent of CD8 positive LCs and/or raw numbers per area of BD or associated underlying submucosa would be worthy of evaluation. The increase of CD8 cells in poorer prognosis cases presumably correlates to an inactive status of these cells.  The demonstration of an increase in density and/or proportion of these cells in the invasive SCC group would provide some indication of the degree of unresponsiveness in these settings.   -> We have added the following sentencesp7&8, l202-208: Using image analysis with Qupath software, the mean density of lymphocytes was at 1503/mm2 +/-231 in the submucoa. Mean density of CD8 positive cells in the submucosa was measured at 1693/mm2+/-334 and 1384/mm2+/-322 lymphocytes in patients with and without a medical history of squamous cell carcinoma respectively (p=0.72). Mean density of CD8 positive cells within the epithelium is 674/mm2+/-251. Mean density of CD8 positive cells in the epithelium was measured at 703/mm2+/-342 and 625/mm2+/-407 lymphocytes in patients with and without a medical history of squamous cell carcinoma respectively (p=1).

Conversely, the dichotomous classification of PDL1 positive and negative cases is common and parallels scoring for invasive lung cancers.  However, data should be presented regarding the statistically significant difference in PDL1 positive BD cases versus the normal cohort (does this cohort include squamous metaplasia without atypia?). It is not too surprising that there are no detectable differences between moderate dysplasias and the high grade group consisting of severe BD and CIS as moderate dysplasias show characteristics that are often closer to high grade than low grade subgroups.  Thus, presenting the data differentiating the BD from the normal tissues should be highlighted. -> p3, l98-100 : We have modified the following sentence “We compared the results of PD-L1 expression obtained in bronchial dysplasia to PD-L1 expression in 30 cases of normal bronchial epithelium and to 20 cases of squamous metaplasia without atypia.”

We have added the paragraph 3.2 (p4, l139-144) to highlight the results differentiating BD from the normal tissues.

While the small size of the study set and the limitations of IHC as the only technique to quantify cell populations are drawbacks, with some major revisions the manuscript could provide some informative observations related to the progression of premalignant lesions to invasive lung cancer.

Reviewer 2 Report

The study demonstrates the PD-L1 and CD8 expression in bronchial dysplasia, the pre-neoplastic lesions for invasive squamous cell carcinoma. Overall, I feel this study is suitable for publication in Diagnostics with a few changes.

  1. In Table 2, PD-L1 expression by dysplastic cells was not significantly correlated with the medical history of squamous cell carcinoma (p=1). To me, the correlation analysis in Table 2 does not support one of the conclusions in the Abstract – PD-L1 expression is an early event in the squamous cell carcinoma carcinogenesis. Please revise the conclusions accordingly.
  2. It would be great to have the patient smoking status and run correlation analysis between smoking and PD-L1 and CD8 expression.
  3. Please add scale bars to Figure 1
  4. There is a typo in the second row of Table 2 – PD-L1 expression by dysplastic cells not dyplastic cells

Author Response

We would like to thank reviewer#2 for her/his comments. We have chosen to resubmit our revised manuscript with the revision suggested by reviewers. We think that these revisions have improved the manuscript and hope you will find it suitable for publication.

The study demonstrates the PD-L1 and CD8 expression in bronchial dysplasia, the pre-neoplastic lesions for invasive squamous cell carcinoma. Overall, I feel this study is suitable for publication in Diagnostics with a few changes.

  1. In Table 2, PD-L1 expression by dysplastic cells was not significantly correlated with the medical history of squamous cell carcinoma (p=1). To me, the correlation analysis in Table 2 does not support one of the conclusions in the Abstract – PD-L1 expression is an early event in the squamous cell carcinoma carcinogenesis. Please revise the conclusions accordingly. -> We have modified the last sentence of the abstract to “It suggests that PD-L1 is expressed in preneoplastic lesions of squamous cell carcinoma.”
  2. It would be great to have the patient smoking status and run correlation analysis between smoking and PD-L1 and CD8 expression.-> We have added the following sentences :
  • P3, l134-135: All patients with dysplasia were smokers with a mean pack-years of 44.3+/-5.5.

P7,l185-186: “All patients with dysplasia were smokers with a mean pack-years of 39.7+/-4.5.” and “The number of pack-years was not related to PD-L1 expression and CD8 positive lymphocytes localization (p=1)”

We have also modified table 2 according to reviewer #3 comments also on smoking status.

  1. Please add scale bars to Figure 1.-> We have added a scale bar for Figure 1
  2. There is a typo in the second row of Table 2 – PD-L1 expression by dysplastic cells not dyplastic cells -> We have modified this error in the second row of Table 2

Reviewer 3 Report

Dear Editor,

Mansucript "Immune escape is an early event in pre-invasive lesions of lung squamous cell carcinoma" by Laville D et al demonstrates important findings about expression of PD-L1 and CD8 in normal bronchial epithelium in comparison with squamous metaplastic and dysplastic epithelium. Their findings suggest the importance of PD-L1 associated mechanism for the development of squamous cell carcinoma in the lungs.

There are some points, which should be addressed more in details:

  1. At the end of Introduction, in aims-section, please rewrite this paragraph, and put it in more or less one sentence, without always repeating We analyzed...
  2. For all study groups it is absolutely necessary to show sex and age distribution but also smoking status!!! These should be of similar distribution; otherwise it is hard to compare the results! Especially the smoking status.
  3. Please comment in discussion the problem of representativity of small samples (biopsies) vs surgical material (heterogeneity of PD-L1 expression)
  4. Under which magnification was PD-L1 evaluation performed (200 or 400?)
  5. Did the authors try to see what happens if cut off for PD-L1 is set at 50%?
  6. Minor English corrections are needed.

Kind regards

Author Response

We would like to thank reviewer#3 for her/his comments. We have chosen to resubmit our revised manuscript with the revision suggested by reviewers. We think that these revisions have improved the manuscript and hope you will find it suitable for publication.

Dear Editor,

Mansucript "Immune escape is an early event in pre-invasive lesions of lung squamous cell carcinoma" by Laville D et al demonstrates important findings about expression of PD-L1 and CD8 in normal bronchial epithelium in comparison with squamous metaplastic and dysplastic epithelium. Their findings suggest the importance of PD-L1 associated mechanism for the development of squamous cell carcinoma in the lungs.

There are some points, which should be addressed more in details:

  1. At the end of Introduction, in aims-section, please rewrite this paragraph, and put it in more or less one sentence, without always repeating We analyzed... -> p2, l72-77 : We modified this sentence to “We analyzed 1/ the relationship between PD-L1 and CD8 expression with the degree of dysplasia, 2/ the relationship between PD-L1 or CD8 expression and medical history of invasive squamous cell carcinoma, 3/ and the proportion of PD-L1 positive lesions of bronchial dysplasia.”
  2. For all study groups it is absolutely necessary to show sex and age distribution but also smoking status!!! These should be of similar distribution; otherwise it is hard to compare the results! Especially the smoking status. -> p6 & p 7 : We have not added sex distribution because most patients (23/24) were males. We have added these informations in Table 2.
  3. Please comment in discussion the problem of representativity of small samples (biopsies) vs surgical material (heterogeneity of PD-L1 expression)-> p9, l277-279 : We have added the following sentence and the according reference at the end of the discussion section “PD-L1 status might not be representative on small samples leading to underestimation of PD-L1 expression on small samples than overestimation [25].”
  4. Under which magnification was PD-L1 evaluation performed (200 or 400?) -> p3, l108: We have modified the first sentence of 2.3 paragraph to “PD-L1 expression was quantified by two observers jointly (DL &FF) evaluating the percentage of neoplastic cells with membranous expression at x200 magnification.”
  5. Did the authors try to see what happens if cut off for PD-L1 is set at 50%? -> We have not included this data in the manuscript because the number of patients in the >50% group is too small to conclude anything.
  6. Minor English corrections are needed. -> We have corrected minor mistakes and typo errors in this manuscript.

Round 2

Reviewer 1 Report

The manuscript is well-improved by the modifications.  One additional change that would be helpful would be to use "intra-lesional" or "intra-epithelial" instead of "intra-tumoral" in tables 1 and 2 and in the text as I believe now invasive carcinomas are analyzed in this study.  Intra-tumoral suggests invasive carcinoma and could create confusion for readers.

Author Response

We would like to thank reviewer #2 for her/his additionnal comments.

We agree with this comments. We have modified intra-tumoral to "intra-epithelial" in tables and in the text.